# High Oxidation Resistance of CVD Graphene-Reinforced Copper Matrix Composites

**DOI:** 10.3390/nano9040498

**Published:** 2019-04-01

**Authors:** Mingliang Wu, Baosen Hou, Shengcheng Shu, Ao Li, Qi Geng, He Li, Yumeng Shi, Minghui Yang, Shiyu Du, Jun-Qiang Wang, Shuzhi Liao, Nan Jiang, Dan Dai, Cheng-Te Lin

**Affiliations:** 1College of Physics and Information Science, Key Laboratory of Low-dimensional Quantum Structures and Quantum Control of the Ministry of Education, Synergetic Innovation Center for Quantum Effects and Applications, Hunan Normal University, Changsha 410081, China; wumingliang@nimte.ac.cn; 2Key Laboratory of Marine Materials and Related Technologies, Zhejiang Key Laboratory of Marine Materials and Protective Technologies, Ningbo Institute of Materials Technology and Engineering (NIMTE), Chinese Academy of Sciences, Ningbo 315201, China; 15184448276@163.com (B.H.); shushengcheng@nimte.ac.cn (S.S.); liao@nimte.ac.cn (A.L.); gengqi@nimte.ac.cn (Q.G.); lihe@nimte.ac.cn (H.L.); jiangnan@nimte.ac.cn (N.J.); 3Center of Materials Science and Optoelectronics Engineering, University of Chinese Academy of Sciences, Beijing 100049, China; jqwang@nimte.ac.cn; 4International Collaborative Laboratory of 2D Materials for Optoelectronics Science and Technology of Ministry of Education, College of Optoelectronic Engineering, Shenzhen University, Shenzhen 518060, China; yumeng.shi@szu.edu.cn; 5Ningbo Institute of Materials Technology and Engineering (NIMTE), Chinese Academy of Sciences, Ningbo 315201, China; myang@nimte.ac.cn (M.Y.); dushiyu@nimte.ac.cn (S.D.); 6Key Laboratory of Magnetic Materials and Devices & Zhejiang Province Key Laboratory of Magnetic Materials and Application Technology, Ningbo Institute of Materials Technology & Engineering, Chinese Academy of Sciences, Ningbo 315201, China

**Keywords:** CVD graphene, graphene/copper composites, oxidation resistance, electrical contact materials

## Abstract

Copper-based materials are common industrial products which have been broadly applied to the fields of powder metallurgy, electrical contact, and heat exchangers, etc. However, the ease of surface oxidation limits the durability and effectiveness of copper-based components. Here, we have developed a powder metallurgy process to fabricate graphene/copper composites using copper powders which were first deposited with graphene layers by thermal chemical vapor deposition (CVD). The graphene/copper composites embedded with an interconnected graphene network was then able to be obtained by vacuum hot-pressing. After thermal oxidation (up to 220 °C) in humid air for several hours, we found that the degree of surface oxidation of our samples was much less than that of their pure Cu counterpart and our samples produced a much smaller increase of interfacial contact resistance when used as electrical contact materials. As a result, our graphene/copper composites showed a significant enhancement of oxidation resistance ability (≈5.6 times) compared to their pure Cu counterpart, thus offering potential applications as novel electrical contact materials.

## 1. Introduction

Copper is one of the most widely used industrial materials in the fields of powder metallurgy [1], electrical contact [2], and heat exchangers [3] due to its excellent electrical conductivity, thermal conductivity, and high workability [4,5,6]. However, in most application cases, the copper workpiece is directly exposed to air, leading to the formation of surface oxide and patina layers, which reduce the reliability and service life of the workpiece [7,8]. In particular, copper workpieces used as electrical contacts are more easily oxidized due to the elevated temperature caused by friction and joule heating during switching [9,10]. In order to avoid surface oxidation, techniques have been adopted, such as the fabrication of corrosion-resistant alloys [11,12], coating with inert metals [13], and cladding with organic layers [14]. However, the inherent problems brought about by the above techniques, such as degradation of thermal and electrical conductivities in the alloy system [12], loss of availability at elevated temperature conditions, and ease-of-abrasion while being coated with organic layers [14], cannot be ignored. Therefore, a long-lasting oxidation resistance enhancement method for copper materials with less negative effects on other physical properties is in high demand.

Graphene, a two-dimensional atomically-thin carbon layer composed of an sp^2^-hybridized lattice, has triggered great research excitement [15] due to its ultralow electrical resistivity (theoretically ≈10^−8^ Ω m) [16], high specific surface area (over 2000 m^2^ g^−1^) [17], and robust chemical inertness [18,19]. In particular, the impermeability of graphene to most gases, liquids, and ions in aqueous solution makes it suitable for barrier applications [20,21,22]. Additionally, the hydrophobic nature of graphene prevents the formation of hydrogen bonds of the coated substrates with physically adsorbed water, resulting in the improvement of corrosion resistance of the substrate surface in humid environments [20]. Hence, so far much work has focused on the use of graphene as an effective nanometer-thick barrier that enables the enhancement of oxidation resistance of the metal surface [22,23,24]. Among various types of graphene and its derivatives, graphene films grown by chemical vapor deposition (CVD) are a promising choice with which to carry out the barrier task due to their low concentration of surface defects, which may increase the possibility of transported molecules contacting the substrate [19,25]. Chen et al. have demonstrated the protective ability of graphene layers grown by CVD to avoid oxidation of the surface of copper foils, both in warm air and in hydrogen peroxide aqueous solution [23]. In addition, Bolotin et al. have reported that graphene films directly formed on copper by catalytic CVD show better antioxidative performance than CVD graphene transferred onto copper, although transferred graphene can also reduce surface oxidation compared to that which is uncoated [26].

The advantages of graphene for barrier applications to protect copper from oxidation and corrosion have been demonstrated [20,23]. However, in reality, a thin graphene layer coated on the surface of metallic components is easily frictionally removed by mechanical action, thus greatly limiting the antioxidative ability of graphene coatings [23]. To solve this problem, the formation of the graphene framework within the bulk copper, rather than within a coating on the surface, provides a potential route to approaching both good protection performance and durability during mechanical operations [26,27,28]. Lee et al. encapsulated copper powders with a thin polymer layer used as a solid carbon source, followed by annealing to obtain graphene-coated copper powders. A vacuum hot-pressing was then performed to produce copper composites incorporated with graphene [29]. However, the high I_D_/I_G_ ratio (≈0.86) in the Raman spectra suggests the graphene layer converted from the polymers is defective. As a result, currently, the fabrication of functional copper composites embedded with low-defect and well-defined graphene frameworks is still a great challenge.

In this paper, a large-scale synthesis process was developed for preparing graphene/copper (Gr/Cu) microparticles by thermal CVD, followed by vacuum hot-pressing to fabricate Gr/Cu composites. Based on the uniform dispersion of low-defect few-layer graphene within the composites, the oxidation resistance of Gr/Cu composites was found to be 5.6 times greater than its pure Cu counterpart.

## 2. Materials and Methods

The production of Gr/Cu powders synthesized by catalytic CVD has been reported in a previous study of ours [19]. High-quality, monodispersed Gr/Cu powders were successfully prepared using a conventional CVD process with the employment of powder spacers. After CVD, Gr/Cu powders were compacted in a graphite mold, followed by hot-pressing at 1000 °C for 30 min with an applied pressure of 35 MPa and a background pressure of <10^−3^ MPa. The surface of the obtained Gr/Cu composites was sequentially polished using 240, 400, 800, 1000, and 2000 mesh alumina waterproof abrasive papers. A pure Cu counterpart was also prepared using the same process for comparison.

The graphene quality was analyzed by a Raman spectrometer (Renishaw plc, Wotton-under-Edge, Gloucestershire, UK) employing a laser wavelength of 532 nm. The crystal structure of the composites was identified by an X-ray diffractometer (XRD, Bruker D8 Advance, Bruker, Karlsruhe, Germany). The sample morphology and elemental compositions were observed and determined by a field emission scanning electron microscope (SEM, QUANTA 250 FEG, FEI, Hillsboro, OR, USA) equipped with an energy dispersive spectral analyzer (EDS).

## 3. Results and Discussion

The fabrication process of the Gr/Cu composites embedded with an interconnected graphene network is schematically illustrated in Figure 1a. In a conventional process for the fabrication of Gr/Cu composites, exfoliated graphene sheets were mixed with copper powders by ball milling or mechanical blending [30,31]. However, the large density difference between graphene (≈2.25 g cm^−3^) and copper (8.9 g cm^−3^) commonly leads to difficulty in obtaining uniform dispersion of graphene in the copper matrix, especially for large scale production. Therefore, in order to solve this problem, our approach was to directly grow graphene layers on the surface of each copper powder by CVD, followed by hot-pressing consolidation to prepare our composites to be 12.6 mm in diameter and 1.9 mm in thickness. Based on our proposed process, the particle size of copper powders (average size ≈50 μm) before and after graphene growth may not change, as presented in Figure 1b, suggesting the convenience of further producing powder metallurgy components. After hot-pressing, compared to that of the Cu counterpart, the grain size of polished Gr/Cu composites was much smaller (see Figure 1c), which is microstructural evidence that graphene forms an interconnected network within the copper matrix and acts as a barrier to inhibit the growth of copper grains at high temperature.

The quality and layer numbers of graphene in the Gr/Cu composites were identified by Raman spectroscopy. Figure 1d shows the typical peaks of graphene corresponding to the D- (≈1347 cm^−1^), G- (≈1580 cm^−1^), and 2D bands (≈2691 cm^−1^) [32], respectively, while no peak was detected in the Raman spectrum of the Cu counterpart. A small D-band can be seen in Figure 1d with an estimated ID/IG ratio of ≈0.19, which is due to the intrinsic metal impurities of copper powders, resulting in the formation of a small amount of amorphous carbon [32]. In addition, according to the I_2D_/_IG_ ratio (≈1.07) and the full-width at half-maximum of the 2D-band (≈50.2), we concluded that the few-layer graphene was synthesized with a layer number of less than 5 [32]. In Figure 1e, only the peaks at 43.4°, 50.5°, and 74.2° corresponding to the (111), (200), and (220) planes of cubic copper (JCPDS No. 04-0836) appear in the XRD pattern of the Gr/Cu composites, which is similar to that seen with the Cu counterpart. No peak for Gr/Cu composites can be found at ≈26°, which is the position of the (002) lattice plane of graphite (JCPDS No. 04-0836), suggesting the atom-thick nature of the synthesized graphene layers. As a result, Gr/Cu composites incorporated with an interconnected graphene network can be reliably made by powder metallurgy.

In order to investigate the improvement of oxidation resistance of the Gr/Cu composites based on the good barrier properties of low-defect graphene, both the Gr/Cu composites and the Cu counterpart were heated in air at 220 °C for 6 h at an ambient humidity of 50 ± 5%. According to previous studies [21], the conversion of the copper surface from bright to dark is associated with the degree of surface oxidation and the thickness of the oxide layer because CuO is black and Cu_2_O is red. Figure 2a presents two samples before and after thermal oxidation taken at the same background brightness of photographs and OM images. Although the color of both as-prepared samples is similar, after treatment we found that the surface color of the Gr/Cu composites was much lighter than that of the counterpart, suggesting a lower degree of surface oxidation of Gr/Cu composites.

The degree of surface oxidation of the samples was quantitatively identified by EDS mapping at an acceleration voltage of 5 kV. At least three spots (about 300 μm × 300 μm) were analyzed to produce the EDS spectra. A typical EDS spectrum of Gr/Cu composites after treatment is shown in Figure 3a; in this spectrum Cu, O, Au, and C elements were detected. The appearance of a weak carbon signal may have originated from the contribution of either graphene or carbon paste to fixing the samples, and the gold peak would have come from a thin deposition layer on the sample surface for improving image quality. As the results of the EDS analysis show in Figure 3a, the similar oxygen content of both as-prepared samples (≈35 wt%) can be attributed to the formation of a native oxide layer on the sample surface. After thermal oxidation, the oxygen content of Gr/Cu composites slightly increased to ≈39 wt%, whereas it increased to ≈47 wt% for the Cu counterpart, demonstrating the superior oxidation resistance of our Gr/Cu composites in atmosphere. In Figure 3b, compared to that of the untreated sample, the surface morphology of Gr/Cu composites after treatment is shown to become rough and textured due to the formation of oxide nanostructures [21,23]. Accordingly, a weak peak can also be found at 36.4° corresponding to the (111) plane of Cu_2_O in the XRD pattern in Figure 3c (JCPDS No. 65-3288).

One of most industrial applications for copper and its alloys/composites is their use as electrical contact materials [2]. Therefore, oxidation resistance ability is a key factor for copper-based electrical contacts when operating in atmosphere. Without it, the interfacial contact resistance (ICR) between two electrodes will significantly increase when an oxide layer is formed on the mating surfaces. In order to identify the extent of oxidation resistance of Gr/Cu composites for potential electrical contact applications, ICR measurement equipment was developed, as schematically shown in Figure 4a. A sample 12.6 mm in diameter and 1.9 mm in thickness was set between two polished copper blocks (indenters) and the total electrical resistance was recorded. Considering that the intrinsic electrical resistivity of our sample, indenters, and electrical leads are small, the measured total electrical resistance came mostly from the ICR, which is as a function of the applied clamping pressure [33]. In Figure 4b, it can be seen that the ICR of the samples initially decreased and then reached a constant value with an increase of the applied clamping pressure. When in a steady-state with a clamping pressure of 30 kgf cm^−2^, the ICR of the as-prepared samples (Gr/Cu composites and Cu counterpart) was similar. However, after thermal oxidation at different temperatures for 6 h, we found that the ICR of the Cu counterpart increased monotonically with an increase in treatment temperature. This increase in ICR can be attributed to the formation of an insulating oxide layer on the sample surface at high temperatures in air. By contrast, there was only a slight increase of ICR in the case of the Gr/Cu composites. The ICR increase of the samples compared to the as-prepared ones (at 30 kgf cm^−2^ clamping pressure) is plotted in Figure 4c. With the incorporation of an interconnected graphene network in the copper matrix, we have clearly demonstrated that our Gr/Cu composites have a better (i.e., 5.6 times higher) oxidation resistance ability than the samples made of pure copper, showing promising performance for their use as novel electrical contact materials.

## 4. Conclusions

In this study, copper-based composites embedded with an interconnected graphene network were fabricated by CVD growth of graphene on the surface of copper powders, followed by vacuum hot-pressing. Due to this characteristic microstructure and the barrier nature of low-defect graphene, the oxidation resistance of the obtained Gr/Cu composites was able to be significantly improved relative to that of the pure Cu counterpart. The ICR measurements indicate that the surface of the Gr/Cu composites was slightly oxidized even after thermal oxidation at 220 °C in atmosphere for several hours, leading to a small ICR increase. Our finding provides a new strategy for the development of copper-based electrical contact materials by modulation of the graphene microstructure within the matrix.

## Figures and Tables

**Figure 1 nanomaterials-09-00498-f001:**
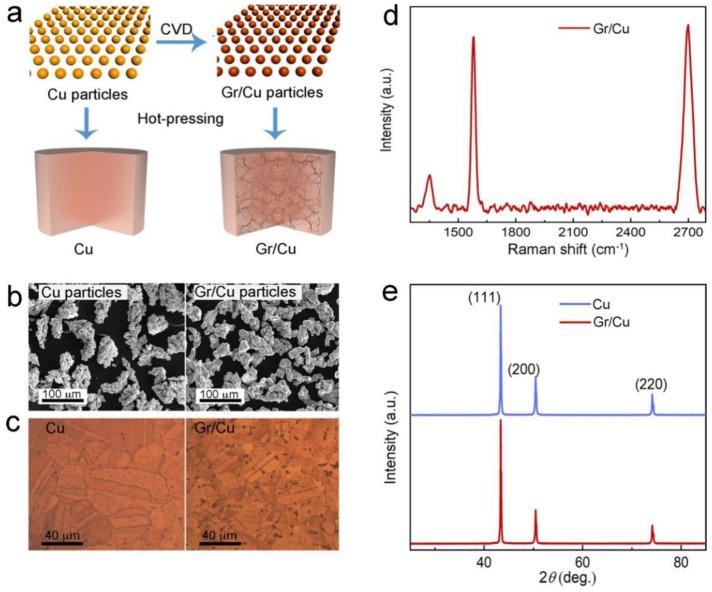
(**a**) Schematic of the fabrication process of Gr/Cu composites with defined graphene microstructure. (**b**) Scanning electron microscopy (SEM) images of Cu powders before and after graphene growth. (**c**) Metallographic images of Cu counterpart and Gr/Cu composites. (**d**) Raman spectrum and (**e**) X-ray diffraction (XRD) pattern of Gr/Cu composites. Legend: CVD, chemical vapor deposition.

**Figure 2 nanomaterials-09-00498-f002:**
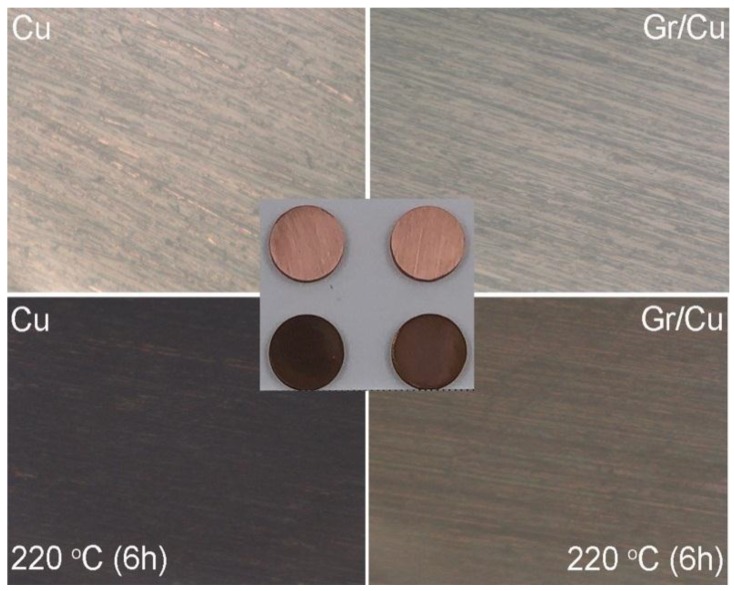
Photographs and OM images of Cu counterpart and Gr/Cu composites before and after thermal oxidation.

**Figure 3 nanomaterials-09-00498-f003:**
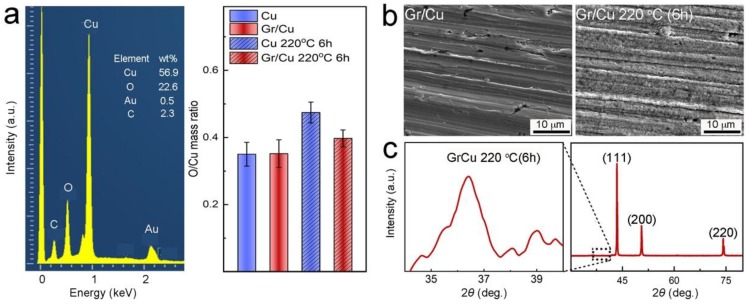
(**a**) A typical energy dispersive spectroscopy (EDS) spectrum of Gr/Cu composites after treatment and oxygen content analysis from EDS measurements. (**b**) SEM images and (**c**) XRD pattern of Gr/Cu composites after treatment.

**Figure 4 nanomaterials-09-00498-f004:**
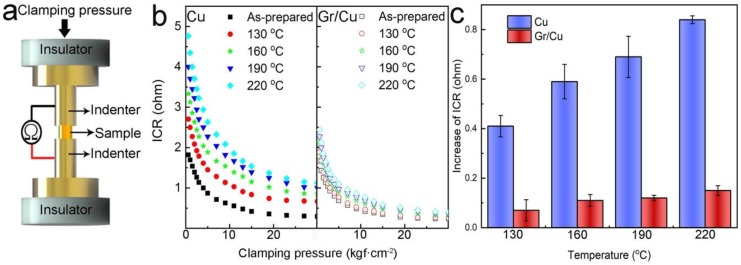
(**a**) Scheme of self-built equipment used for interfacial contact resistance (ICR) measurements. (**b**) ICR as a function of the clamping pressure for Cu counterpart and Gr/Cu composites thermally treated at different temperatures and (**c**) the corresponding ICR increases.

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
