# Peer review of "High Oxidation Resistance of CVD Graphene-Reinforced Copper Matrix Composites"

_nanomaterials, 2019, doi:10.3390/nano9040498_

Reviewer 1 Report

The work shows an interesting approach to solve the oxidation issue of Cu, it does provide some useful insight for a further improvement.

I would recommend to check the english as few mispelling and rewriting might be needed.

line 46 joule heat should be changed in Joule heating

line 52-54 please rephrase, resistivity should be changed with resistance

line 58 in particularly should be changed "in particular, specifically"

line 75-78 please rephrase the initial part of the sentece.

line 80 grapheme should be graphene

line 104-105 sentence not so clear

line 109 I assume the micron symbol is missing

Fig 2 , (c) appears twice

 In addition to this I would like to point out something else:

- a brief summary of the synthesis process would be useful even if referenced 

- the author state that no peak was observed for the raman spectrum of pure Cu, such spectrum is not shown

- the author state that no XRD peak was observe at 26 deg, It seems that Fig 1e start from 30 deg, please extend the x axis

- EDS is not a good quantitative method to calculate Oxygen content, how many different spot were analyzed to produce the table in Fig 2?. It should be clearly stated and they should be several to have a minimum of reliability.

- The measurement of ICR is not clearly described, a ohm-meter was used? which one? Each value comes from a single measurement or it is an average of several values?

Author Response

Dear Editor and Reviewers,

We are very grateful to your and the reviewers’ critical comments and thoughtful suggestions. Based on these comments and suggestions, we have made careful revision on the original manuscript (Manuscript ID: nanomaterials-451881). A revised manuscript has been submitted, in which the modified sections are marked in red. Thank you and reviewers again, who made great contribution to improve our paper. We responded point by point to the reviewer comments as listed below, along with a clear indication of the location of the revision.

Question 1: The work shows an interesting approach to solve the oxidation issue of Cu, it does provide some useful insight for a further improvement.

I would recommend to check the English as few misspelling and rewriting might be needed.

line 46 joule heat should be changed in Joule heating

line 52-54 please rephrase, resistivity should be changed with resistance

line 58 in particularly should be changed "in particular, specifically"

line 75-78 please rephrase the initialpart of the sentence.

line 80 grapheme should be graphene

line 104-105 sentence not so clear

line 109 I assume the micron symbol is missing

Fig 2, (c) appears twice.

Reply:Thanks for your suggestion. The mistakes you listed has been corrected. Changes in the revised manuscript were highlighted in red. 

Question 2: A brief summary of the synthesis process would be useful even if referenced.

Reply:Thank you for your suggestion,a brief summary has been added in the manuscript (line89- 90).

Question 3: The author state that no peak was observed for the Raman spectrum of pure Cu, such spectrum is not shown.

Reply:Thank you for your suggestion, The Raman spectra of Cu and Gr/Cu are exhibited in Figure R1, showing that peak was observed for the Raman spectrum of pure Cu.

Figure R1: Raman spectra of Cu and Gr/Cu

Question 4: The author state that no XRD peak was observe at 26 deg, it seems that Fig 1e start from 30 deg, please extend the x axis.

Reply:Thanks for your suggestion, we have extended the x axis of XRD pattern in Figure 1e.                                                                                         

Question 5: EDS is not a good quantitative method to calculate Oxygen content, how many different spot were analyzed to produce the table in Fig 2?. It should be clearly stated and they should be several to have a minimum of reliability.

Reply: Thanks for your comment. At least three spots (about 300 m´300 mm) were analyzed to produce the table in Figure 3a. We also produced the statement in the manuscript. 

Question 6: The measurement of ICR is not clearly described, a ohm-meter was used? which one? Each value comes from a single measurement or it is an average of several values?

Reply:As shown in Figure 4a, the ICR values is tested by the self-built equipment. We measured the total electrical resistance (including our sample, indenters, electrical leads). Considering that the intrinsic electrical resistivity of our sample, indenters, and electrical leads are small, the measured total electrical resistance comes mostly from the ICR. We approximate the measured electrical resistance as ICR values, which is as a function of the applied clamping pressure. Each ICR value comes from the average of at least three measurement values.

We appreciate for Editor/Reviewers’ warm work earnestly, and hope that the correction will meet with approval. The manuscript has been overall checked, and the changes marked in red font one by one. We hope that these revisions are sufficient to make our manuscript acceptable for publication in Nanomaterials. If you have any question about this paper, please do not hesitate to contact me.

Yours sincerely,

Cheng-Te Lin 

Ningbo Institute of Material Technology & Engineering, Chinese Academy of Sciences

Reviewer 2 Report

I find the manuscript very interesting.

My only suggestion to improve the article is to change Figure 2 in taht way that the material structures are made in a larger format as separate figures. Improving the quality of images presenting the structure of composites will certainly facilitate the reception of the description.

Author Response

Dear Editor and Reviewers,

We are very grateful to your and the reviewers’ critical comments and thoughtful suggestions. Based on these comments and suggestions, we have made careful revision on the original manuscript (Manuscript ID nanomaterials-451881). A revised manuscript has been submitted, in which the modified sections are marked in red. Thank you and reviewers again, who made great contribution to improve our paper. We responded point by point to the reviewer comments as listed below, along with a clear indication of the location of the revision.

Question 1: My only suggestion to improve the article is to change Figure 2 in that way that the material structures are made in a larger format as separate figures. Improving the quality of images presenting the structure of composites will certainly facilitate the reception of the description.

Reply: Thanks for your suggestion. We have changed Figure 2 that the material structures have been made in a larger format as separate figures (Figure 2 and Figure 3 in revised manuscript).

We appreciate for Editor/Reviewers’ warm work earnestly, and hope that the correction will meet with approval. The manuscript has been overall checked, and the changes marked in red font one by one. We hope that these revisions are sufficient to make our manuscript acceptable for publication in Nanomaterials. If you have any question about this paper, please do not hesitate to contact me.

Yours sincerely,

Cheng-Te Lin 

Ningbo Institute of Material Technology & Engineering, Chinese Academy of Sciences

Reviewer 3 Report

Figure 3, text on page 5, line 173:

"a clamping force of 30 kgf cm-2" physically means pressure. So either units or naming is wrong.

Author Response

Dear Editor and Reviewers,

We are very grateful to your and the reviewers’ critical comments and thoughtful suggestions. Based on these comments and suggestions, we have made careful revision on the original manuscript (Manuscript ID nanomaterials-451881). A revised manuscript has been submitted, in which the modified sections are marked in red. Thank you and reviewers again, who made great contribution to improve our paper. We responded point by point to the reviewer comments as listed below, along with a clear indication of the location of the revision.

Question 1: "a clamping force of 30 kgf cm-2" physically means pressure. So either units or naming is wrong.

Reply: Thanks for your suggestion. “Clamping force” have been replaced by “clamping pressure”.

We appreciate for Editor/Reviewers’ warm work earnestly, and hope that the correction will meet with approval. The manuscript has been overall checked, and the changes marked in red font one by one. We hope that these revisions are sufficient to make our manuscript acceptable for publication in Nanomaterials. If you have any question about this paper, please do not hesitate to contact me.

Yours sincerely,

Cheng-Te Lin 

Ningbo Institute of Material Technology & Engineering, Chinese Academy of Sciences
